# Seeking the Building Blocks of Visual Imagery and Creativity in a Cognitively Inspired Neural Network

Shekoofeh Hedayati, Roger Beaty, Brad Wyble

Department of Psychology
The Pennsylvania State University
University Park, PA 16802
shokoufeh.hed@gmail.com

## Abstract

How do we imagine visual objects and combine them to create new forms? To answer this question, we need to explore the cognitive, computational and neural mechanisms underlying imagery and creativity. The body of research on deep learning models with creative behaviors is growing. However, in this paper we suggest that the complexity of such models and their training sets is an impediment to using them as tools to understand human aspects of creativity. We propose using simpler models, inspired by neural and cognitive mechanisms, that are trained with smaller data sets. We show that a standard deep learning architecture can demonstrate imagery by generating shape/color combinations using only symbolic codes as input. However, generating a new combination that was not experienced by the model was not possible. We discuss the limitations of such models, and explain how creativity could be embedded by incorporating memory mechanisms to combine the network's output into new combinations and use that as new training data.

## 1   Introduction

For millennia philosophers have been interested in our ability to conjure memories and objects as a functional component of our visual system. Visual imagery allows us to re-experience remembered content and may enable us to mentally manipulate that information in a format that is similar to the original input from the eyes (Kosslyn, Thompson & Gianis 2006).

Computational models can provide a functional intuition about how information is transformed from sensory input, memories and visual knowledge, into an experienced visual form. However, recent models that allege to demonstrate "creative" behavior (e.g. Ramesh at al., 2021) are too complex to be understood as a model for human imagery and creativity. It is also difficult to determine the boundaries of creativity in such models. For instance, models like GPT-3 (Brown al., 2020), CLIP (Radford et al., 2021) and DALL-E (Ramesh at al., 2021) have been trained on datasets that are so large that it is impossible to understand the full scope of the examples they have been exposed to. While the generative output of these models is impressive, it is difficult to determine if such output is an interpolated sample that provides intermediate variation from the training set. Moreover, models with large training sets generate harmful and inappropriate contents which creates further barriers for using them as inspiration for studying creativity (Birhane et al., 2021). The emphasis in artificial intelligence research is often to build ever larger models with the idea that models that are large and trained on sufficiently massive data sets will approach human levels. However, the larger the systems get, the harder it will be to understand how they work, to know whether they are truly creative, and to use them as a lens to study human creativity. With this regard, truly creative means being able to

3rd Workshop on Shared Visual Representations in Human and Machine Intelligence (SVRHM 2021) of the Neural Information Processing Systems (NeurIPS) conference, Virtual.

utilize learned knowledge while combining elements in a way that generates a novel form. Our goal is to move in the opposite direction, with a cognitively and biologically plausible model that uses simpler machinery.

Bearing in mind the importance of avoiding inappropriate inference from artificial networks to neural systems (Guest & Martin, 2021), we are not arguing that our model is similar to human imagery and creativity. Rather, we are using this model to develop more accurate intuitions about how high dimensional systems can operate, which in turn allows us to think more clearly when developing theories. The proposed model in this paper is based on the Memory of Latent Representations (MLR) model developed by Hedayati, O'Donnell and Wyble (2021). The MLR model uses a fully connected neural network that shows how hierarchical visual knowledge generates and retrieves visual stimuli and stores these into working memory tokens. Due to its generative feature and its biological relevance to the visual system (i.e., resembling the hierarchical structure of visual ventral stream with more generic representations at early levels and more compressed representations at higher levels; Figure1A), we modified this model to explore visual imagery and creativity. Finally, previous research has shown the critical role of working memory in creative tasks (Benedeck et al., 2014).Therefore another advantage of using MLR is that it has an embedded working memory component that can, if necessary, be used to combine information and generate new forms. It is important to bear in mind that the common notion of creativity is generating something that is novel and useful. In this sense we are focusing on novelty, because usefulness is outside of the scope of our model. In the the proposed model, the visual knowledge system was trained on a colorized MNIST (LeCun, 1998) dataset, and we will be testing 1) generating color-shape combinations in the absence of a direct visual stimuli by using symbolic codes as demonstrated by imagery 2) generating novel shape-color combinations as a form of visual creativity.

## 2 Method and Results

### 2.1 Model Components

Our model consists of a modified variational autoencoder representing distinct features (dfVAE; Kingma & Welling, 2013) as used in the MLR model, and two multiple layer perceptron (MLP) networks. The former approximates the visual knowledge that is trained on a colorized version of the MNIST dataset, whereas the latter converts categorical labels into latent representations. We trained the model on MNIST consisting of 60,000 of handwritten digits ranging from 0-9, which was colorized by 10 prototype colors with minor variations and had the dimension of 28x28x3 pixels. The training and testing data were chosen based on the original MNIST dataset and the model's parameters were identical to the dfVAE model proposed by Hedayati et al.(2021) with the addition of the label networks. We implemented the model with Pytorch 1.4.0 and CUDA v.10.0. The model was trained using a single Nvidia 2080 Ti GPU with a batch size of 100. The full training took approximately 1-2 hours.The code is available at: https://github.com/Shekoo93/Imagery.

### 2.2 dfVAE

This part was directly taken from the MLR model. The encoder and the decoder of the dfVAE are similar to the original VAE proposed by Kingma and Welling (2013), except for the bottleneck layer (i.e., mid-layer) that was separated to represent shape and color information disjointedly by using two distinct objective functions. This disentanglement allowed for reconstructing a specific feature (shape or color) in the output. Layers of the model have L1=256, L2=128, bottleneckshape=4, bottleneckcolor=4, L4 = 128 and L5=256 neurons. The VAE learns to compress and decode the visual stimuli by generating them in the output, and comparing them with the input image to accomplish the training in an unsupervised fashion. As shown in Figure 1A, we can see layers of the VAE coarsely correspond to the anatomical architecture of the visual ventral stream.

### 2.3 Label network

For the purpose of generating images in the absence of a direct visual stimulus, we added two MLPs each of which had 10 input neurons, 7 neurons in the middle layer and terminate on the 4-neuron shape or color map respectively. The label network started with a categorical (one-hot) coded representation and learned to map those codes onto representations in the shape and color maps

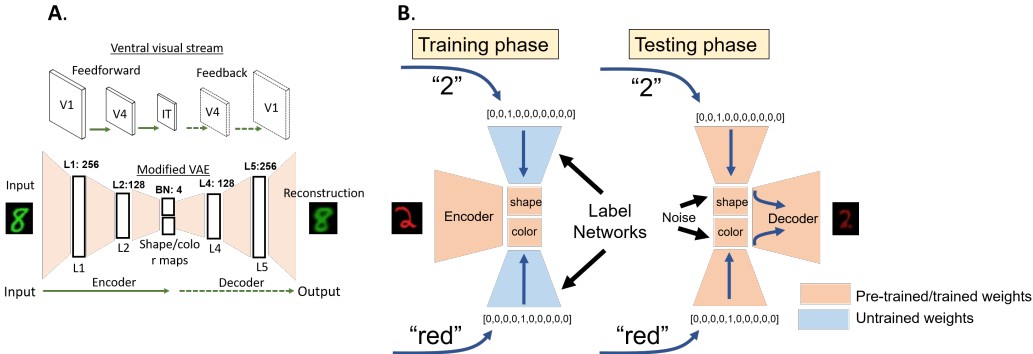

Figure 1: Panel A: the architecture of the dfVAE with separate shape and color maps in the bottleneck (BN). Each layer of the dfVAE corresponds to a region of the ventral visual stream. The dfVAE reconstructs the input image in the output layer. Panel B: The proposed model's architecture with label networks attached to a pre-trained dfVAE. Initial training of the label networks is shown on the left (stage 1), and after training the label networks are able to generate arbitrary shape-color combinations in the absence of visual inputs (stage 2). The binding pool, which stores and retrieve from working memory is not shown.

that were generated by the dfVAE's encoder based on the visual stimuli.Training the label network involved activating a color/shape label while presenting the corresponding input image (Figure1B). Gradient descent minimized the mean squared error between the activation generated by the labels and the activation generated by the corresponding image. After the training phase was completed, our model could create combinations of handwritten digits with different colors using the one-hot codes which we take as an analog of visual imagery.

## 2.4 Imagery Simulation 1: Familiar combinations

Figure 2 (Panel A) shows how imagining a "purple two" changes as we add increasing noise to the shape/color maps. The label networks first activate a representation in the shape and color maps that collectively produce a purple 2 at the output. Increasing amounts of noise are then added to these latent representations which are passed through the decoder to generate the images. This simulation demonstrates that random perturbation of a representation does not generate new shapes or colors.

## 2.5 Imagery Simulation 2: Novel combinations

In simulation 1, the model was trained on all 100 combinations of the 10 shapes and 10 colors. To test the model's ability to generate novel shape-color combinations, in Simulation 2 we trained the model on red 0-4 and green 5-9 MNIST digits and tested its ability to generate red 5-9 or green 0-4 With 150 attempts. This model was unable to generate a combination outside of its training set. Figure 2B shows 50 examples of noisy imaginations of red 2, green 5 and green 2. The model generates only combinations of the red and green digits that it was trained on. In the lower third panel the label network tries to reconstruct a specific combination that it was not trained on (green 2) and the model is unable to do so.Also see Figure3A (top) for a TSNE map illustration of the shape map and its comparison with a TSNE map (Figure 3A, bottom) for a model that was trained on red and green digits with complete overlap.

## 2.6 Imagery Simulation 3: Novel combinations with overlapped training

In the previous simulation, the two sets of training stimuli were completely disjoint (red 0-4, green 5-9). It is possible that partial overlap of color-digits in the training set would allow the dfVAE decoder to generate more flexible representations that enable the reconstruction of novel digit-color pairings. To evaluate this possibility, a new model was trained with the following sets: red 0-3 and green 2-5, such that digits 2 and 3 were presented in both red and green. As shown in Figure 2C, even with this new training set, red 5 could not be generated but green 5 could be generated.
To explore the latent space of the model that failed to generate a green 2 combination, we trained

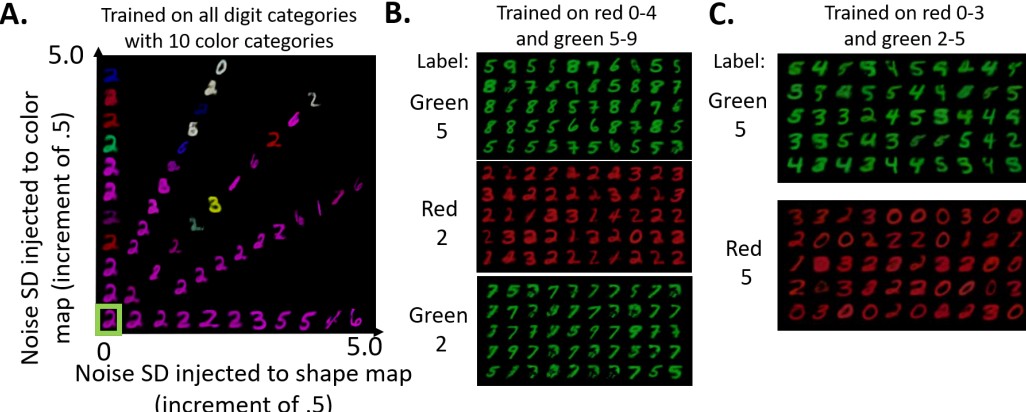

Figure 2: The image within the green rectangle is the reconstructed "purple 2" in the absence of noise on the model that was trained on all combinations of 10 digits and colors. Panel B, Imagined images of the model that was trained on red 0-4 and green 5-9. The 50 images shown for each digit-color combination are the result of combining the label network's output with Gaussian noise (SD=1) added to the shape map. On the bottom, the label network tries and fails to generate a green 2. In Panel C, for the third simulation, even a model with overlapping shape and color combinations is unable to generate representations that are outside of its set of trained combinations.

classifiers on shape and color maps respectively. The average accuracy of decoding color from the shape map for 5 models was 75.8% (SE= 1.24, chance=50%) when the model was trained on non-overlapping digits (red 0-4 and green 5-9). On the other hand, the average accuracy of decoding color from the shape map was 60.6% (SE=1.46) when the model was trained on all combinations of red and green digits. Figure 3A shows TSNE maps for shape and color categories for different training sets. The clusters within the maps show that red and green representations are more entangled with how entangled digits are based on their colors, and that clusters are more spread out when combinations have complete overlap.

### 2.7 Imagery Simulation 4: Novel combinations with diverse overlapped training

In this manipulation, we aimed to increase the latent space generalization by training the dfVAE on MNIST and Fashion-MNIST (f-MNIST; Xiao et al., 2017) examples represented in green and red, except that "2" was available only in red and "6" was available only in green. The results indicated that the dfVAE could not combine the shape and color to generate new forms such as a "green 2" or "red 6". Figure 3B indicates reconstruction examples directly from dfVAE.

## 3 Discussion

MLR provides a simple, biologically plausible model of visual imagery that combines a hierarchical visual representation with a highly flexible working memory system. The training set uses clearly delineated combinations that we can use to study the building blocks of creativity in neural systems, such as imagery.

The model exhibits imagery in the sense that it can generate specific examples of targeted combinations of digits and colors based on labels provided by other putative cognitive system (e.g. executive control or perhaps mechanisms associated with mind-wandering). For example if asked to create a ""red 2" and provided with noise, it is able to generate a variety of images that are a red-2 in the absence of actual visual input. When noise is increased the imagined digits can change from one category to the next, but exhibit very little in the way of intermediate forms. Moreover, the model exhibits no ability to generate novel combinations of shape and color, such that it cannot generate a green 2 if it was trained on red 2's and green 5's. Even when the green and 2 latent representations have been activated by the label networks, the decoder has not learned how to combine these specific representations at the output, so the failure to combine the features is in the decoder. Note that all of

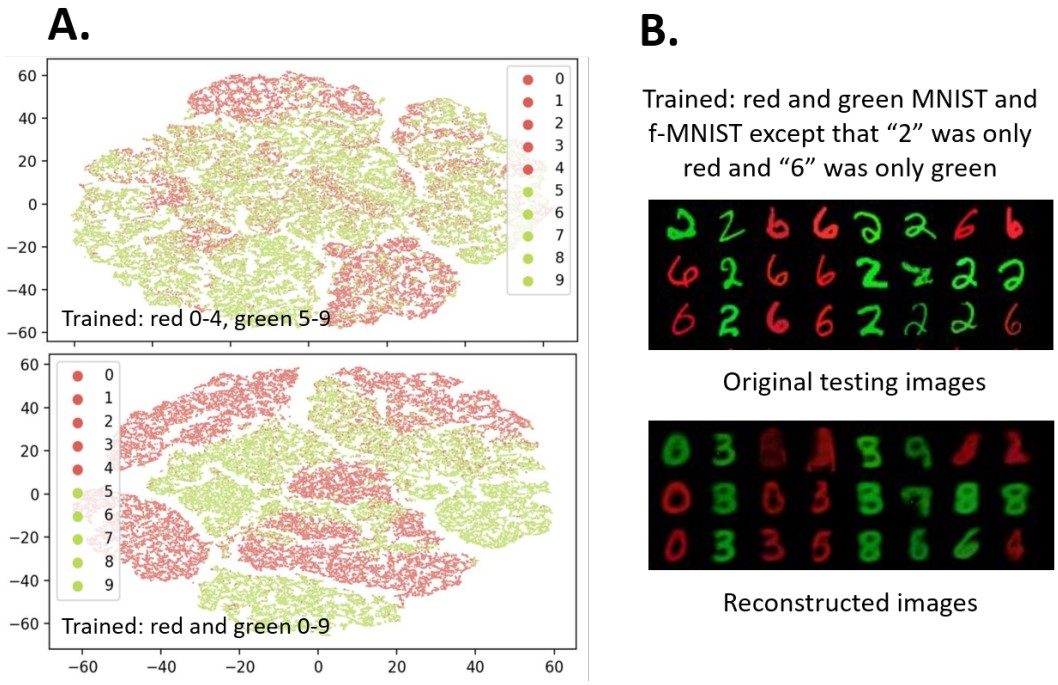

**A.**

Trained: red 0-4, green 5-9

Trained: red and green 0-9

**B.**

Trained: red and green MNIST and
f-MNIST except that "2" was only
red and "6" was only green

Original testing images

Reconstructed images

Figure 3: TSNE illustration for the shape map in panel A, when the model was trained on red 0-4 and green 5-9 (up) vs. when it was trained on red and green digits from 0 to 9 (bottom). The figure shows TSNE reconstruction of the latent representation of the digits red 0-4 and green 5-9 for both models. Panel B illustrates examples of the direct reconstructions from dfVAE for red 2's and green 6's, when the model was trained on red and green MNIST and f-MNIST, but "2" was presented in only red and "6" was presented only in green. The model is unable to generate green "2" and red "6"

the green pixels required for a 2 can be found in the green 7 and 5, which means that the model has learned to generate green pixels along the bottom of the image, and yet, the closest approximation of a green 2 is a green 7 (Figure 2B). This suggests that that there is no pathway connecting the latent representation of 2 in the shape map to green pixels in the output.

The simulations indicate that at least the comparatively simple VAE architecture is unable to generate truly novel combinations. While it remains an open question as to whether more complex models such as GPT-3, CLIP and DALL-E are genuinely creative, the example provided here suggests that we should develop more explicit means to evaluate the extent to which they can generate novel content.

### 3.1 Adding Mechanisms for learned creativity

Since MLR is inspired by cognitive and neurally plausible mechanisms of visual representation and working memory, it gives us a framework to explore possible mechanisms to induce creativity by using memory. Creativity in humans is often a process of memory retrieval combined with recombination (Dietrich, 2004; Feldhusen 2010) and the MLR model provides a conceptual platform to explore these ideas. Furthermore, because autoencoders like a VAE generate output that matches the format of the input, any output can be used to generate new training examples for the input space. This allows the model's expertise to grow to include a superset of combinations of real-world stimuli it has been trained on through self-generated replay. This requires additional mechanisms that can be explored via models that have cognitive components such as working or episodic memory. For instance, any given image can be generated and briefly stored in memory at the decoder's output to be combined with other pieces of output. The combination could then be used as new data for training. As a simple example of this, if the model is trained on a horizontal line and a vertical line but not a '+' shape, it would be unable to simulate the imagination of a '+', since there is no representation of this more complex shape in any of its latent spaces. However, it would be possible for the model to retrieve a memory of the two lines and superimpose them on the output space, and then use this

newly generated '+' shape as training data, allowing it to learn representations of the '+' in all of its latent spaces, as if that symbol had been present in its original training set.

More complex compositional learning and generation algorithms (e.g. Lake et al., 2011) could be used to achieve more complex representational combinations of shape as well. This form of providing additional training to the model based on recombination of its existing representations might be akin to the act of dreaming, in which hippocampal areas of the brain are often highly active and thought to be retrieving information. The advantage of using imagined visual forms as training data for the model is that these new forms become part of the permanent representational architecture of the system and thereby allow it to respond rapidly and efficiently to those forms in the future, despite having never experienced them before.

Other forms of representational manipulation and combination are also possible in a cognitively inspired system. For example, color information is thought to be represented in a way that allows it to be projected across surfaces even when it is not physically present (Figure 4).

Similar mechanisms of color projection could conceivably be used to generate new color-shape combinations at the output layer. For example, if a given color is first projected from the decoder to the output space, and this is followed by a shape, the visual system might map the color onto the shape to create a new combination of shape and color. While it is premature to stipulate exactly what neural circuits would be responsible for such a transform, it is within the scope of functions that the intricate circuitry of visual cortex could accommodate. As suggested above, these novel shape/color combinations could then be projected back through the model as a form of training, allowing them to be ensconced into the encoder, decoder and shape/color maps.

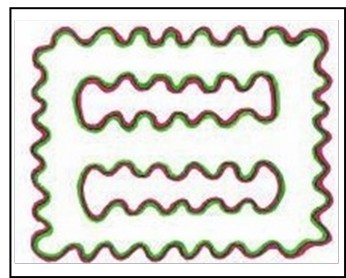

Figure 4: Watercolor illusion, in which the visual system expands the colors of red and green to create an illusory percept of colored surfaces.(Brenna, Brelstaff & Spillman 2001).

Such imagination of novel combinations thus seems within the scope of a model that has both memory and generative output. A better question than how does creativity occur is perhaps how a creative model would guide its creative imagination so that such retraining did not include all possible combinations of stimuli which might overload the model's representational space with conjunctions that never occur in the real world and thus are not helpful.

Another important point to consider is that there could be adaptive value in having an imaginative output at the earliest levels of vision (I.e. the output layer of MLR, that we consider to correspond to visual cortex which receives extensive feedback connections from higher level brain areas) to be constrained during waking behavior to imagine only stimuli that correspond to actual objects that have been experienced. A visual system without such controls might be overly prone to hallucinations that interfere with the ability to perceive. In such a case, the aspect of creative imagery that generates entirely novel forms in a visual format could be the province of generative visual workspaces that are not associated with perception. This would be unlike the framing of the MLR model, which assumes that the pixel level representations of both input and output correspond to early visual areas. It is notable that visual imagery of remembered items tends to activate early visual areas (Kosslyn  Alpert 1993;Kosslyn, Ganis & Thompson 2001) and this would support the MLR interpretation, but it is possible that other brain areas are involved in the imagination of novel forms and this is supported by studies that fail to find visual cortex activation when using imagery to solve a problem (Knauff, Kassubek, Mulack & Greenlee 2000).

In conclusion, we have explored the ability of a modified VAE to create novel combinations of highly familiar visual features and found that it is not able to do so, even when the training sets for the two features (color and shape) overlap in the training set. However, when coupled with memory systems and other aspects of representational interplay within visual areas, it is easy to theorize about how an autoencoder could generate new forms and then use those as training data to expand the model's representational repertoire in a way that would be more consistent with some of the simplest aspects of creativity.

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
