# OpenReview forum: "Seeking the Building Blocks of Visual Imagery and Creativity in a Cognitively Inspired Neural Network"
_NeurIPS.cc/2021/Workshop/SVRHM — SVRHM 2021 Poster_

### Official Review · Reviewer_NmT4 · 2021-10-26
**Review of "Seeking the Building Blocks of Visual Imagery'**

**Rating:** 6
**Confidence:** 3

**Review:**


The goal of this project is to model creativity with a simple model that incorporated biological-plausible design components. Specifically, the authors use a small VAE that learns separate representations for color and shape. They train it on MNIST digits with different colors applied. They find that a) the model does not appear to learn any representations intermediate to the learned letters (i.e. no shape intermediate to 2 and 3), b) a model trained on a subset of the possible color-number combinations is not able to retrieve all possible color-number combinations.

Strengths

- the goal of building a model that can flexibly re-combine different features (i.e. shape and color) is an interesting one, and the approach of a VAE that learns two separate latent spaces on the same stimuli is clever
- It is important to show that a simple model if this sort is limited in the kinds of novel re-combinations it can achieve. It is informative to see that this kind of approach is not able to learn representations that are intermediate to trained representations, and that it cannot generate novel combinations of known features.

There were a couple weaknesses that prevented me from fully engaging with the results.
1) Many of the motivation for this model were not spelled out. The authors use a simpler model to “develop more accurate intuitions about how high dimensional systems can operate”, but they do not spell out how this model will get you there. It would help the reader if they could say what principles they believe are embodied in the architecture of the network, why this model might be informative for this question.

2) The authors state that the model is biologically plausible, but this is not well supported. They suggest that the encoder/decoder stage map onto feed forward/feedback mechanisms in the brain, but there is no reason to suppose that this actually corresponds between humans and machines. Also, they suggest that the layers of the model map onto steps of the visual processing hierarchy, but there’s no evidence that the representations in the model layer match the representations in the human hierarchy.

I see two components that are inspired by human visual processing: hierarchical steps, and a separation between color and shape. However, it’s important to be clear that this loose inspiration doesn’t mean that the model can map onto human visual processing. Overall, they need to 1) be more specific about what components are biologically inspired, and 2) be more moderate in the claim that the VAE maps on to visual hierarchy that closely (e.g. Figure 1).

3) One clear takeaway is that a VAE, with these stimuli is not able to learn intermediate representations. What do the authors suggest we conclude about this? Did this fail because a) this is impossible to learn, b) a VAE is the wrong architecture for this tasks, or c) there wasn’t enough variety in the stimulus set? Likewise, what does it mean that the model couldn’t generate novel combinations? Does this mean that the color and shape representations in the middle of the VAE are not leaned independently? Or are those independent, but the decoder is not able to recombine them? Overall, I expected more of an analysis of why this architecture failed at the two tasks, so that I could understand what to take away from these experiments. The Discussion talked a lot about future directions, but didn’t clearly state what we’ve learned from this model.

Minor comments

Lines 47-48 were hard to parse - didn’t understand what the goals were going to be until I read further below
Figure 2 would be much clearer with more labels - Label panels A/B/C with what they are simulating; within panel B, label which combinations were in the training set and which weren’t.
Line 133 - do you have a citation for that?

Typos

Line 16 - don’t need “The” at the beginning of the sentence “The visual imagery”
Line 50 By convention, should just be “Model Components”
Line 113 “putative cognitive system[S]”
Line 139: “for instance, given memory [,] any given
Line 193 is a word missing?: “combinations of highly [?] visual features”
Figure 1B as L1 and L2 indicated, but not L4 and L5. My intuition is that if one is indicated, so should the other, for consistency

---

### Official Review · Reviewer_4J3T · 2021-10-30
**Interesting direction with potential to be useful to the research community although results are inconclusive.**

**Rating:** 6
**Confidence:** 5

**Review:**

Easy to follow and concise. Well written.

"an interpolated sample that provides intermediate variation from the training set": Philosophically, I would argue that human creativity can be viewed like that too. Creativity in general is so subjective. One could perhaps think of human creativity as a mere interpolation of prior experiences - visual, acoustic, olfactory, multimodal sensory experiences in general - a person has, that get conjured and combined in various ways to produce new things. Are we assessing them as "creative" because we don't all have the same prior experiences or, even if we do, we don't interpolate them in the same way? That being said, accepting this definition of "creativity", models like GPT, CLIP, Dall-E can be considered creative even if they interpolate data from their training sets. This sounds in agreement with the first introductory sentence. Perhaps authors could provide how creativity is defined in order to better understand how something can be considered or assessed as "creative"? Ln 133 touches upon the definition of creativity but that was closer to the end. It would have been helpful to have read the paper having in mind already how authors define human creativity.

Ln 33: in a similar vein as above, what does "truly creative" mean?

Took a look at the code as well to confirm and it looks like authors are training labels and images completely independently. I would find it rather difficult for the architecture to learn to associate labels with their visual counterparts unless they are somehow trained simultaneously in some fashion. Authors could consider trying training both simultaneously with some contrastive loss optimization. Recent work following a similar approach for video prediction with simultaneous training of visual input and labels: https://openreview.net/pdf?id=LdVQGdXkkG

Great analysis in section 3.1. Good ideas there: superimposition of shapes to generate an unseen one (+ sign example). Very much like the inspiration from the human visual cortex. Also, many related works on concept grounding look into generalization as a means to expand a model's knowledge set, however this work looks into generalization as a means of studying or imitating human imagination and creativity which is very refreshing.

Very interesting direction, however the difficulty is that results seem inconclusive: the approach did not work for creating new combinations of shape/color however that does not mean that other approaches using mVAE and MLR wouldn't work. As mentioned above, suggest trying simultaneous learning with some form of fusion of the concept representations. Despite the inconclusiveness of results however, this is a direction that has potential to be useful to the research community, therefore there is value to be discussed in SVRHM 2021.

---

### Official Review · Reviewer_1uQ7 · 2021-10-31
**Review of "Seeking the Building Blocks of Visual Imagery and Creativity in a Cognitively Inspired Neural Network"**

**Rating:** 6
**Confidence:** 2

**Review:**

This paper studies the capacity for simple neural networks to combine visual objects to create new ones, which they use to understand creativity and visual imagery.

Pros:

- The topic is highly interesting.

- The paper's motivation for using simple neural networks over large models (like GPT-3, CLIP, or DALL-E) is compelling.

Cons:

- On the one hand, the paper emphasizes that this this model is advantageous because of its biological plausibility (although this is just stated and not explained). One the other hand, the paper explains that the goal is to build intuitions about how neural networks generate new combinations of data, without claiming that the model intends to capture human imagery. If so, then why is biological plausibility relevant? Some clarification would be helpful.

- Simulations are interpreted in a cognitively-rich manner: the paper assumes that the simulations tap into some form of visual imagery or creativity, but this lacks justification. It seems to me that the paper is simply testing how neural networks extrapolate data in simple settings. It seems then like a large percentage of work un deep learning could count as testing some form of creativity or imagery. It is not clear how the simulations in the paper tackle something unique that departs from the vast literature studying extrapolation (without calling it imagery or creativity).

- The results are somewhat minimal, compared to average contribution in NeurIPS workshops (and in previous SVRHM workshops).

---

### Official Review · Reviewer_ncod · 2021-11-01
**A step towards constraining how visual working memory gives rise to visual creativity**

**Rating:** 4
**Confidence:** 3

**Review:**

Introduction/Motivation:
This paper aims to help understand how latent visual representations learned from a set of visual samples may (not) facilitate the generation of newer patterns. It comes at an opportune time in the wake of many heavyweight networks like GPT-3, CLIP, and DALL-E which, though have impressive performance in generating seemingly novel outputs, have billions of parameters and have been trained on very large and often proprietary datasets. This prevents us from understanding the mechanism by which creativity manifests in these models. Any step in this direction will help us not only understand how to make machines more creative but may also constrain the space of possible mechanisms by which humans demonstrate creative thinking.

Main Takeaways:
A simple variational autoencoder (VAE)-based model cannot be used to generate combinations of shape and color that were not present in the training space. In other words, the VAE's learning is specific to the space of samples it has been exposed to.

Pros:
1. The motives of the paper are clearly written and commendable.
2. The experiment design and the results are straightforward to understand.

Comments:
This paper provides a negative result that an earlier model (MLR) that can learn latent representations (memories) of visual inputs and use them to reconstruct stimulus samples, is unable to generate samples that need the network to combine representations across different latent spaces. While this result is interesting, it may not be able to tell us much about visual creativity for the following reasons:
1. the link between working memory and visual creativity is tenuous (see Remoli, T. C., & Santos, F. H., 2017 for a review). This result would have been more striking if a clear connection between these constructs had been established. Given that this relationship is not so straightforward, it is not clear why we should expect this network to demonstrate visual creativity in the first place.
2. the paper did not test all possible manipulations. Only the impact of the level of noise and adding a slight overlap between the two spaces are the manipulations explored. In other words, how do we know that the manipulations presented here are sufficiently strong to get the network to generate novel visual samples? Some kind of visualization/analysis of the latest space representations would be helpful here. For example, we might see this pattern if the samples belonging to red and green classes were two clouds in a high dimensional space separated by a large distance, and the different numbers are smaller clouds that are separated by a smaller distance. If this is the case we may ask, for example: what is the order of magnitude of the within-class variance/across class variance? If this value is very low, it might be relevant for great classification performance but may be detrimental to the creation of novel stimuli (this idea is similar to overfitting). Without these kinds of analyses, it is not possible to understand why we do not see the expected results.
3. Nothing specific was done to 'induce' the network to produce a novel combination. One might argue that if a human was shown a large number of samples of numbers such that 0-4 were red and 5-9 were green, and then the human is asked to generate some numbers based on what they saw, the humans may default to drawing 0-4 in red and 5-9 in green. If the human is specifically be instructed that they can use the color of their choice, then it is possible that they come up with novel combinations, but in this case, it is unclear what would be an equivalent manipulation for the model. This kind of justification becomes important for a negative result - if the model was able to generate samples with the expected combination of shape and color, it is clear that the network is working as intended.

While the issues topic and issues raised in this paper are very interesting and pertinent, due to the reasons listed above, the paper is not acceptable in its current form.

---

### Decision · Program_Chairs · 2021-11-02

Accept (Poster)